# Vertical Section Observation of the Solid Flow in a Blast Furnace with a Cutting Method

**Yuanxiang Lu [1], Zeyi Jiang [1,2,*], Xinru Zhang [1,3], Jingsong Wang [4] and Xinxin Zhang [1,2]**

[1]  School of Energy and Environmental Engineering, University of Science and Technology Beijing,
    Beijing 100083, China; luyuanxiang2008@163.com (Y.L.); xinruzhang@ustb.edu.cn (X.Z.);
    xxzhang@ustb.edu.cn (X.Z.)
[2]  Beijing Key Laboratory of Energy Saving and Emission Reduction for Metallurgical Industry,
    Beijing 100083, China
[3]  Beijing Engineering Research Center of Energy Saving and Environmental Protection, Beijing 100083, China
[4]  State Key Laboratory of Advanced Metallurgy, Metallurgical and Ecological Engineering, University of
    Science and Technology Beijing, Beijing 100086, China; wangjingsong@ustb.edu.cn
*   Correspondence: zyjiang@ustb.edu.cn; Tel.: +86-010-6233-2741

**Abstract:** The solid flow plays an important role in blast furnace (BF) ironmaking. In the paper, the descending behavior of solid flow in BFs was investigated by a cold experimental BF model and numerical simulation via the discrete element method (DEM). To eliminate the flat wall effect on the structure of solid flow in lab observations, a cutting method was developed to observe the vertical section of the solid flow by inserting a transparent plate into the experimental BF model. Both the experimental and numerical results indicated that plug flow is the main solid flow pattern in the upper and middle zones of BFs during burden descending. Meanwhile, a slight convergence flow and a deadman zone form at the lower part of the bosh. In addition, the boundary between the plug flow and convergence flow in BFs was determined by analyzing the velocity of the burden in vertical directions and the Wilcox–Swailes coefficient ($U_{ws}$). The results indicated that the $U_{ws}$ can be defined as a critical value to determine the solid flow patterns. When $U_{ws} \geq 0.65$, the plug flow is dominant. When $U_{ws} < 0.65$, the convergence flow is dominant. The findings may have important implications to understand the structure of the solid flow in BFs.

**Keywords:** blast furnace; solid flow; cold experiment; direct element method; Wilcox–Swailes coefficient

## 1. Introduction

Blast furnaces (BFs) are complex metallurgical reactors that produce pig iron. During the ironmaking process for BFs, the layered coke and ore particles are charged into the top of the BF. The hot gases are injected into the raceway from tuyeres at the bottom of the BF. Then, the coke particles descend to the raceway and the hearth, and are gasified and combusted mainly in the lower part of the BF. Meanwhile, the ore pellets are reduced during the process of burden descending and gradually become small and soft until they melt to liquid iron. In such a complicated multiphase chemical reaction system, the descending behaviors of coke and ore in BFs directly affects the gas flow distribution, heat–mass transfer, and gas–solid reactions in the BF, which all play a significant role in achieving a smooth operation of the BF. Therefore, it is necessary and important to understand the descending behaviors of coke and ore particles in BFs.

In fundamental aspects, the descending of coke and ore particles in BFs is a typical solid flow. Due to the difficulty of experiments, the solid flow in BFs has been extensively studied by various mathematical models in the past decades, as reviewed by Yagi et al. [1], Dong et al. [2],

Ariyama et al. [3], and Kuang et al. [4]. In recent years, with the development in computer technology, numerical approaches (i.e., computational fluid dynamics (CFD) and the discrete element method (DEM)) have been increasingly adopted as important research tools to investigate the solid flow in BFs. However, it is difficult to track the detailed properties of single particles from the flow field at the grain scale by CFD. The DEM was first proposed by Cundall and Strack [5] in 1979, then rapidly developed by many scholars as a result of its significant advantages in micromechanics of granular materials. To date, many researchers have studied the influences of various factors on the structure of solid flow in BFs by the DEM method, including the discharging velocity and particle shapes [6], burden layer structure [7–9], molten slag trickle flow [10,11], segregation behavior [12,13], shaft-injected gas distribution [14], the softening and melting behaviors of ferrous burdens [15], the air pressure drop [16], and the flow and wall stress [17]. The DEM has been recognized as an effective method to study the fundamental behavior of solid flow in BFs, and most of the studies have adopted the corresponding simplified conditions and models to reduce the calculation time.

Some scholars have also studied more comprehensive models. Adema et al. [18] compared three types of BF geometry (slot models and a pie-slice) with different particle shapes, and concluded that the geometry used should be carefully chosen as it has a very large influence on the solid flow. Ping [19] et al. evaluated different burden descent models under four charging patterns. These models could predict the positions and shapes of different timelines in BFs, and the results showed that the C/O charging pattern can influence the shape of the cohesive zone and the deadman. Fu et al. [20] proposed two models, i.e., the geometric profile (GP) model and the potential flow (PF) model, to consider the non-uniform descending speed of the burden. These models can obtain the descending speed with different C/O ratios and can be applied for online prediction of operating blast furnaces. Using a numerical fluid–solid coupled method, Xu et al. [21] and Yang et al. [11] simulated the fluid and solid phases to study their interaction, and Hou et al. [22] established a quasi-steady virtual experimental BF model that considers the operation and energy efficiency of a BF. These numerical models reveal the main profile of burden structure combined with the experimental results [23–26], and the cold experimental results in the laboratory can provide the basis for the numerical simulation of the hot state. To date, many findings have indicated that the solid flow in BFs can be divided into four characteristic flow regions, i.e., plug flow, stagnant, funnel flow, and quasi-stagnant zones. In addition, Wright et al. [25] analyzed the solid flow regions by comparing the 3D model with slot models. They found that the stagnant zone in the slot model is smaller than that in the 3D model due to the wall effect, and the slot model may not fully describe the solid flow behavior. On the basis of these studies, Yang et al. [27] compared three types of models, i.e., full 3D models, slot models, and sector models, and proved that slot models cannot describe the anisotropic solid flow in the tangential direction as a result of the wall effect. The sector model, which was a more reliable simplified model, should be used in the future studies. Up to now, many researchers have found that it is necessary to eliminate the wall effect during the numerical study of the solid flow in BFs. However, few experiments have been developed to evaluate and verify the influence of the wall effect on the solid flow in BFs.

To address this gap, herein, a cold experimental BF model was established to analyze the descending behavior of solid flow in BFs, i.e., a 3D half-circle BF (180°) model, which was made of transparent acrylic material. To eliminate the wall effect, a novel cutting method for the burden was developed to observe internal structure of the solid flow in BFs. Furthermore, two numerical DEM sector models were used to verify the experimental models and examine the characteristics of solid flow under different burden layer conditions, such as full coke, layered coke/ore pellet, and mixed coke/ore pellet. In addition, based on the cutting method, the solid flow patterns in BFs were explored. The findings may have important implications to understand the structure of the solid flow in BFs.

## 2. Materials and Methods

### 2.1. Experimental Design

The experiments of solid flow in blast furnaces are often conducted in the laboratories, where the experimental results of the burden structure can be recorded directly by high-speed cameras. Generally, the experimental furnace body is a semicircular structure. There are two walls on the periphery of such a furnace body. One is a semicircular peripheral wall and the other one is a flat wall. The observation surface is the flat wall, and the results are recorded from this surface.

However, due to the observation of the burden structure from the flat wall, there is a physical impact on the solid flow. Compared to the full furnace model without the influence of the flat wall, the half furnace model may have some potential errors on the structure of the solid flow. But it is difficult to observe the internal structure of the burden in the full furnace model. In order to eliminate the impact of the flat wall on solid flow as much as possible, an experimental method is developed in this paper. On the basis of a half BF model, a further exploration of the internal burden structure can be studied. In this way, it can not only observe the descent process of the burden, but also ensure the reliability of the experimental results. This part is arranged in the Appendix A in order to describe the feasibility of this method.

### 2.2. Experiment Setup

A 3D experimental platform of solid flow in BFs was designed and setup through the above experimental idea. The schematic of the cold experimental BF model is illustrated in Figure 1. The BF model was composed of a BF body, a storage bin, a charging device (inlet), and a discharging device which could control the flow rate of solids in the BF using a spiral discharger. The geometry parameters of the BF body model are listed in Table 1. In order to observe the structure of solid flow, an experimental BF model (i.e., a 3D half-circle BF model made of transparent acrylic material) was established. It should be noted that, the experimental BF models were designed with the scale of 1:15 based on the geometric dimensions and operating conditions of a commercial BF (inner volume 125 m$^3$ with 8 tuyeres). Generally, during the ironmaking process of BFs, the coke combustion in the raceway is considered as the main driving force of burden descending. Here, in the cold experimental BF model, the discharge process of particles below the tuyere is regarded as the consumption process of coke combustion. The melting of iron ore in the cohesive zone is ignored.

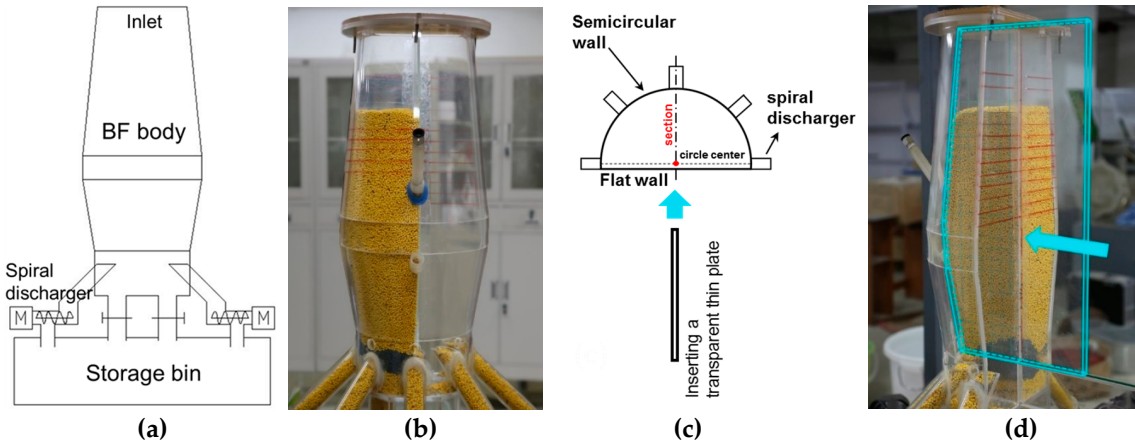

**Figure 1.** The 3D half-circle blast furnace (BF) model. (**a**) Schematic of the 3D experimental BF model. (**b**) The image of the 3D half-circle experiment BF model. (**c**) Schematic of inserting a transparent plate into the 3D half-circle experiment BF model. (**d**) The image of the vertical section of solid flow obtained by the cutting method in the 3D half-circle experiment BF model.

**Table 1.** Dimensions of the experimental model.

| Parameters | Value |
| --- | --- |
| Diameter of hearth, m | 0.213 |
| Diameter of belly, m | 0.267 |
| Diameter of throat, m | 0.200 |
| Height of hearth, m | 0.213 |
| Height of bosh, m | 0.160 |
| Height of belly, m | 0.054 |
| Height of shaft, m | 0.333 |
| Height of throat, m | 0.100 |

*2.3. Particle Properties*

In the experiment, two different cylindrical polyethylene particles were used as the burden particles and tracer particles, which could be used to display the position of burden motion and the distribution of descending velocity. Specifically, in the 3D half-circle BF, the yellow particles and the small dark blue particles were used as the burden particles and tracer particles, respectively. During the experiment, the timelines were formed by these tracer particles, which could be used to analyze the structure of solid flow. It should be noted that, all of these cylindrical polyethylene particles had a repose angle of 40°, because the repose angle of coke generally ranged from 35° to 45°. In addition, these particles had a diameter of 3–5 mm, a real density of 910 kg/m$^3$, a burden porosity of 0.35, and an elastic modulus of 1.07 GPa. The diameter of the tracer particles was 2.5mm. The static friction coefficient between the polyethylene particles and the outer wall of acrylic (i.e., $f_{p\text{-}w}$) is 0.156. The static friction coefficient between polyethylene particles (i.e., $f_{p\text{-}p}$) is 0.21.

*2.4. Experimental Procedure*

Prior to the experiment, the particle descending speed was determined. Generally, considering the particle Froude similarity, the velocities of burden can be calculated by the Froude number [25]:

$$Fr_{\mathrm{s}} = \frac{\rho_{\mathrm{g}}}{\rho_{\mathrm{P}} - \rho_{\mathrm{g}}} \cdot \frac{u_{\mathrm{s}}^2}{g d_{\mathrm{P}}} \tag{1}$$

where $\rho_{\mathrm{g}}$ and $\rho_{\mathrm{P}}$ are the densities of the gas and the solid, respectively. $d_{\mathrm{P}}$ is the equivalent diameter of the particle. $u_s$ is the particle descending velocity at the furnace throat. The modified $Fr_{\mathrm{s}}$ relates the inertial forces acting on the solid and gas phases. According to the operational data of the BF, the particle descending speed in the model was controlled at $3.45 \times 10^{-4}$ m/s.

Before the experiment, a verification experiment was carried out to test the reliability of this cutting method as shown in the Appendix A. In the experiment, firstly, the 3D half-circle BF was filled up with the yellow polyethylene particles. Then, the continuous charging and discharging system was initiated. The first layer of the tracer particles were evenly charged from the top of the experimental BF model. Subsequently, the burden particles (yellow) and tracer particles (dark blue and black) were fed into the BF model alternately. The tracer particles were charged into the BF model at intervals of 3 min, thus forming several thin tracer layers acting as timelines. The charging and discharging system was shut off when the first layer of the tracer particles reached the outlets. By this time, a frozen burden body was formed, which could be used to study the structure of the solid flow.

Furthermore, to eliminate the flat wall effect, a cutting method was developed to observe the internal structure of the solid flow in the frozen burden body. In the cutting method, as shown in Figure 1c, a transparent plate was vertically inserted into the flat wall of the 3D half-circle BF. Then, the solid particles on the other side of the transparent plate were removed, as shown in Figure 1d. After that, the structure of the solid flow on the section in the BF model could be photographed. It should be noted that, as shown in Figure 1c, the shape of the half-circle model was not a true semicircle, but a lengthened semicircle. Specifically, a half furnace plus a lengthened part just met the

span of the five tuyeres, as shown in Figure 1b,c, and the edge of the flat wall just matched the two tuyeres on each end. This design ensured that the burden lines above the tuyeres could be observed from the flat wall, and the internal structure above the tuyere on the section could be observed after the cutting method. The transparent plate inserted into the BF model was along the direction of the reserved smooth grooves, and at the top of the model there was a top cover with a smooth groove, as shown in Figure 1d. Under the action of instantaneous thrust, the transparent plate could cut the particles in the half furnace into "two parts" through two smooth grooves on the top and bottom, and the burden lines were not affected too much by this cutting method.

## 2.5. Discrete Element Method

The DEM method by EDEM commercial software (EDEM$^{TM}$, version 2018, DEM solutions, Edinburgh, England) was adopted to simulate the solid flow in the experimental BF model. Since the BF possesses an axisymmetric structure with eight tuyeres on the lower part of the wall, two 3D sector models of a one-eighth circle (45° as shown in Figure 2) were constructed by considering the efficiency and accuracy of calculation. As shown in Figure 2, in the sector model 1, the wall effect was eliminated (i.e., each $f_{p-w}$ is set as zero). Meanwhile, in the sector model 2, the wal -effect was considered (i.e., one of $f_{p-w}$ was set as 0.156). Additionally, in the DEM simulation, the internal angle of the sector was adjusted to a round arc of 5 mm diameter as shown by the cross-section wall in Figure 2, which can weaken the friction effect of the acute angle.

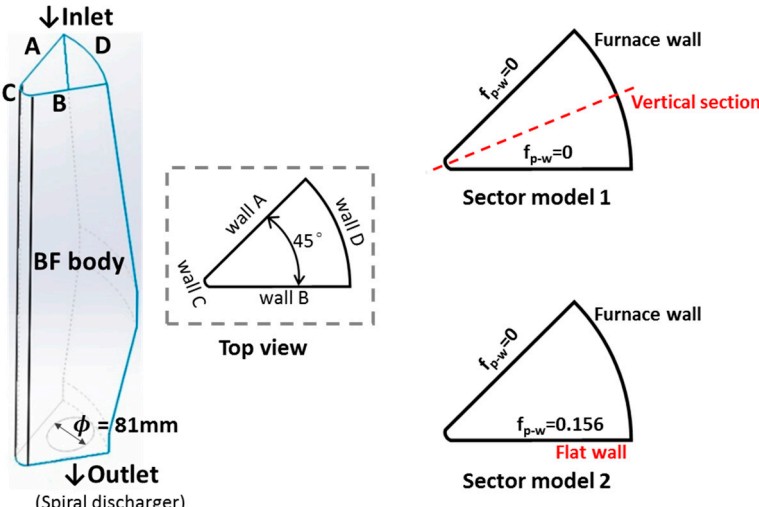

**Figure 2.** Direct element method (DEM) geometry of the two BF sector models.

Generally, the motions of a particle mainly include the translational and rotational motions, which satisfy Newton's second law and the laws of rotation, respectively. The motion of a particle which is pursued by a multi-body interaction based on the soft sphere approximation can be decided by the contact forces from the geometry and particles. The Hertz–Mindlin [28] model including a spring and dashpot was used in DEM to calculate the contact forces. The trajectory of a particle is obtained by considering the translational and rotational motions of a particle [14]. In detail, the governing equations for a particle *i* can be expressed as follows:

$$m_i \frac{dv_i}{dt} = \sum_j \left( F_{cn,ij} + F_{dn,ij} + F_{ct,ij} + F_{dt,ij} \right) + m_i g \tag{2}$$

$$I_i \frac{d\omega_i}{dt} = \sum_j \left( M_{t,ij} + M_{r,ij} \right) \tag{3}$$

where $v_i$ and $\omega_i$ denote the translational velocity (m/s) and rotational velocity (rad/s) of the particle *i*, respectively. $m_i$ is the particle mass in kg/m$^3$. $I_i$ is the moment of inertia of the particle in kg·m$^2$,

which is given by $I_i = 0.4m_i R^2$. The forces include the gravitational force, $m_i g$, and the contact forces between the particles and particle walls. The contact forces and the damping forces in the normal and tangential directions involved are: $F_{cn,ij}$, $F_{ct,ij}$, $F_{dn,ij}$, and $F_{dt,ij}$ (in units of N), respectively. The torque acting on particle $i$ are: $M_{t,ij}$, which causes particle $i$ to rotate by the tangential force, ( in units of N·m) and $M_{r,ij}$, so called the "rolling friction torque", which slows down the relative rotation between particles by the normal force (in units of N·m). The forces and torques used in the model are listed in Table 2.

**Table 2.** Forces and torques acting on particles $i$.

| Forces and Torques | Symbols | Equations |
|---|---|---|
| Normal contact force | $F_{cn,ij}$ | $-2/3 S_n |\delta_n| n$ |
| Normal damping force | $F_{dn,ij}$ | $-2\sqrt{5/6}\beta\sqrt{S_n m^*} v_{n,ij}$ |
| Tangential contact force | $F_{ct,ij}$ | $-S_t |\delta_t| t$ |
| Tangential damping force | $F_{dt,ij}$ | $-2\sqrt{5/6}\beta\sqrt{S_t m^*} v_{t,ij}$ |
| Coulomb friction force | $F_{t,ij}$ | $-\mu_s \left| F_{cn,ij} + F_{dn,ij} \right| t$ |
| Torque by tangential forces | $M_{t,ij}$ | $R^* n \times (F_{ct,ij} + F_{dt,ij})$ |
| Rolling friction torque | $M_{r,ij}$ | $-\mu_r \left| F_{cn,ij} + F_{dn,ij} \right| R \hat{\omega}_i$ |

Notes: $S_n = 2E^*\sqrt{R^*|\delta_n|}$, $n = \frac{\delta_n}{|\delta_n|}$, $\frac{1}{m^*} = \frac{1}{m_i} + \frac{1}{m_j}$, $\beta = \frac{\ln e}{\sqrt{\ln^2 e + \pi^2}}$, $S_t = 8G^*\sqrt{R^*\delta_n}$, $t = \frac{\delta_n}{|\delta_n|}$, $\frac{1}{E^*} = \frac{1-v_i^2}{E_i} + \frac{1-v_j^2}{E_j}$, $\frac{1}{R^*} = \frac{1}{|R_i|} + \frac{1}{|R_j|}$, $\frac{1}{G^*} = \frac{2(1+v_i)(1-v_i^2)}{E_i} + \frac{2(1+v_j)(1-v_j^2)}{E_j}$, $v_{n,ij} = (v_{ij} \cdot n) \cdot n$, $v_{t,ij} = (v_{ij} \cdot t) \cdot t$, and $\hat{\omega}_i = \frac{\omega_i}{|\omega_i|}$, $v_{ij} = v_j - v_i + \omega_j \times R_j - \omega_i \times R_i$. $E^*$, $\delta_n$, $m^*$, $R^*$, $e$, $G^*$, and $S_t$ mean the equivalent Young's modulus, normal amount of overlap, equivalent mass, equivalent radius of the particles, coefficient of restitution, equivalent shear modulus, and tangential stiffness of particles, respectively.

In the DEM simulation, the motion of each particle was traced, and the collision force between the particles was calculated. Equations (2) and (3) were used to record and calculate the velocity of the particles, and thus the motion characteristics of the solid flow were simulated. Specifically, the parameters of the particles in the simulation were shown in Table 3.

**Table 3.** Parameters of the particles in the DEM simulation.

| Parameters | Sector Model 1 | Sector Model 2 |
|---|---|---|
| Particle shape | Spherical | Spherical |
| Particle motion state | Moving bed | Moving bed |
| Particle diameter, mm | 2.5 (c), 1.25 (o) | 2.5 (polyethylene) |
| Particle density, kg/m$^3$ | 1100 (c), 4000 (o) | 910 (polyethylene) |
| Wall density, kg/m$^3$ | 7600 (furnace wall) | 1200 (acrylic) |
| Time step, s | $1 \times 10^{-4}$ | $1 \times 10^{-4}$ |
| Total number | Variable | 35,000 |
| Poisson's ratio | 0.21 (c), 0.24 (o) | 0.49 (polyethylene) |
| Shear modulus, Pa | 1e + 07 | 1e + 07 |
| Coefficient of restitution | 0.3 | 0.3 |
| Coefficient of interparticle static friction | 0.63 (c-c), 0.4 (c-o), 0.32 (o-o) | 0.21 |
| Coefficient of interparticle rolling friction | 0.05 | 0.05 |
| Coefficient of static friction (p - wall A) | 0 | 0 |
| Coefficient of static friction (p - wall B) | 0 | 0 |
| Coefficient of static friction (p - wall C) | 0 | 0.156 |
| Coefficient of static friction (p - wall D) | 0.56 (c-w), 0.31 (o-w) | 0.156 |
| Coefficient of rolling friction (p - w) | 0.05 | 0.05 |

(c: coke particle; o: ore pellet; p: particle; w: wall)

## 3. Results

### 3.1. Vertical Section of the Solid Flow in the Experimental BF Model

To eliminate the flat wall effect, the vertical section of the solid flow was observed by inserting a transparent plate into the experimental BF model using the cutting method. Figure 3 shows the distributions of burden particles and tracer particles in the 3D half-circle BF model observed through the outer wall (flat wall) and the section, in which nine timelines represent the structure of the solid flow.

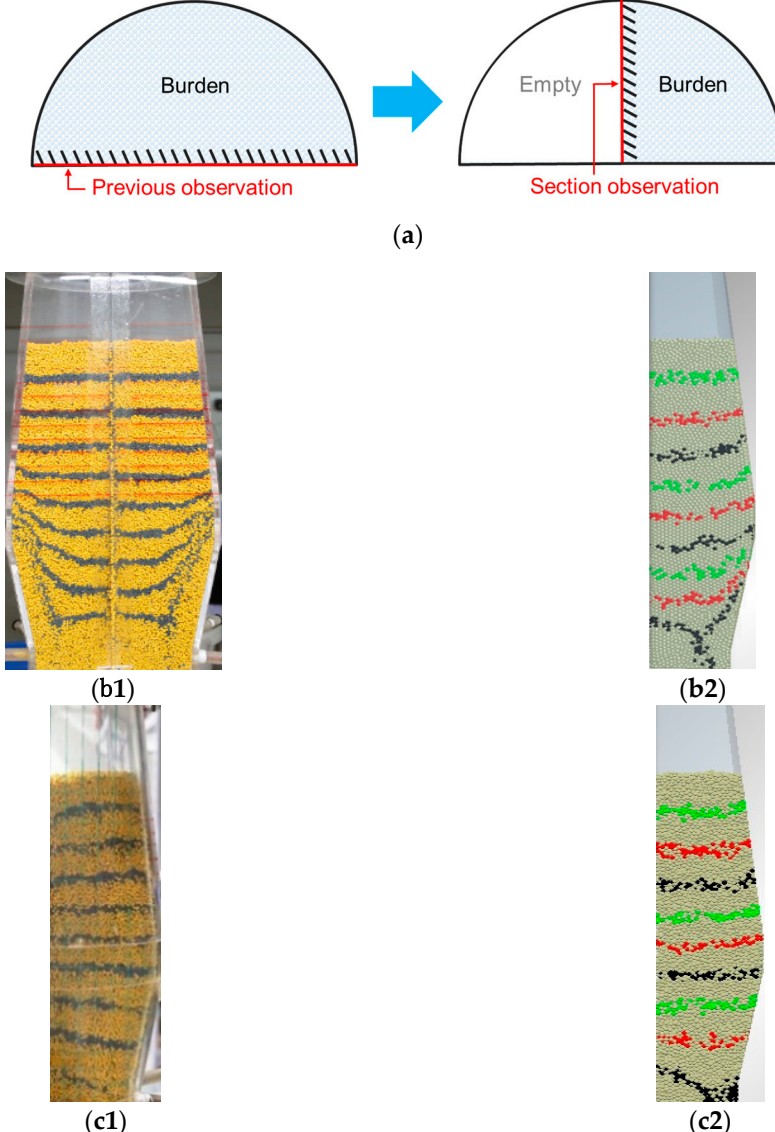

**Figure 3.** (**a**) The section observation by the cutting method in the 3D half-circle BF model. The distributions of burden particles and tracer particles observed through the flat wall (**b1**) and the internal surface after the cutting method (**c1**) in the experiment. Similarly, the flat wall (**b2**) and the internal surface (**c2**) simulated by sector model 2 and sector model 1, respectively.

Figure 3a shows the section observation by the cutting method. The distributions of burden particles and tracer particles in the 3D half-circle BF model were observed through the vertical section after cutting. As shown in Figure 3b, the solid flow in the bosh region, which was observed from the flat wall, appears as an obvious "W"-shaped convergence flow. Furthermore, in order to further study

the solid flow patterns in the BF, the vertical section of the solid flow in the 3D half-circle BF model was observed. Figure 3c1 shows the distributions of burden particles and tracer particles in the 3D half-circle BF model observed through the vertical section after cutting. Clearly, there is a significant difference in the timelines of tracer particles with the results shown in Figure 3b1,c1. Meanwhile, as shown in Figure 3c, the timelines which were observed from the vertical section of solid flow after cutting show approximate straight lines in the shaft area and oblique lines in the bosh area. Compared with the literature [25], there is a longer bosh in this work. The result of the flat wall shows a larger quasi-stagnation zone in the bosh, which illustrates that the shape and length of the bosh can affect the shape of this quasi-stagnation zone. In the hearth region, the experimental result in Figure 3c1 shows that more inclined lines exist in this region, which is different from the results in the literature (the curved lines) [25]. Evidently, the plug flow zone can be extended to the middle of the bosh area. As for the phenomena shown in Figure 3b1, it is likely that, due to the 90° angle between the flat and semicircular walls, the increase in particle velocity at the center and the decrease in particle velocity near the wall led to a long "W" shape in the gradually narrowed bosh region. In addition, it can be found that a vertebral-body deadman exists at the center of the hearth region, and thus, the W shape was obvious at the bottom of the BF model. However, as for the phenomena shown in Figure 3c1, it can be found that when the wall effect was eliminated using the cutting method, the plug flow zone covers most of the area in the BF, which agrees well with the observation shown in the 3D full-circle BF model [27].

Similarly, Figure 3c2 shows an image obtained through the vertical section of the solid flow in the 3D half-circle BF model simulated by sector model 1 via DEM. Evidently, by comparing with the results shown in Figure 3b2, it can be found that, when the wall effect was eliminated, the plug flow is dominant in the BF. Accordingly, the results shown in Figure 3 indicate that only the vertical section of the solid flow can reflect the actual situation of the solid flow due to the elimination of the wall effect. Therefore, in future experiments and simulations, the vertical section of the solid flow should be used to analyze the characteristics of the plug and convergence flows, which is consistent with the literature [25,26]. The experimental cutting method can solve the problem of the wall effect on the burden descending in shaft furnaces, which might have important implications for various industrial applications [24–27].

### 3.2. Vertical Section of Solid Flow Under Different Burden Layer Conditions

To further understand the solid flow behavior under different burden layer conditions, the solid flows for three types of burden layers were simulated by the sector model 1 via DEM, including the full coke, layered coke/ore pellet, and mixed coke/ore pellet. The physical properties of the coke and ore particles are shown in Table 3. Figure 4a–c shows the images of the vertical section of the solid flow, when the burdens are the full coke, layered coke/ore pellet, and mixed coke/ore pellet, respectively. Evidently, the solid flow behaviors under these three types of burden layers are almost the same, i.e., in the middle and upper part of the BF, the plug flow is dominant, whereas, the convergence flow appears below the middle of the bosh.

It is worth noting that the vertical section of solid flow shown in Figure 4a is similar to that shown in Figure 3c2, indicating that the formation of the timelines in the BF was less affected by the properties of the particles [23,24]. However, it can be found that, after adding ore pellets into the BF models, the timelines in the bosh became straighter. Meanwhile, comparing with the vertical section of solid flow shown in Figure 4b,c, we found that a larger quasi-stagnant zone appeared in Figure 4a. It is likely that ore pellets have smaller scale and smaller roughness than coke particles, and therefore the effect of mixing the coke and ore pellets is becoming more and more obvious in the bosh under gravity. In addition, the relatively poor rolling characteristics of the pure coke pellets may be balanced by the mixing process of the coke and ore pellets, which makes the quasi-stagnant zone further smaller. Nevertheless, the vertical sections of solid flow under different burden layer conditions, as shown in Figure 4, all indicate that the plug flow is the main solid flow pattern in the upper and middle

zones during burden descending, whereas a slight convergence flow and a deadman zone form at the lower part of the bosh. Therefore, when analyzing the heat–mass transfer and metallurgical reactions in the BF, the plug flow should be used to model the solid phase, which agrees well with the literature [29]. The results can provide important information for building the one-dimensional model and the two-dimensional model used to simulate the movement and reaction of coke/ore in BFs.

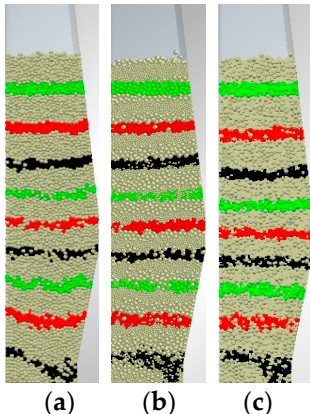

**Figure 4.** Internal wall of solid flow under three burden layer conditions simulated by the sector model 1 via DEM, i.e., full coke (**a**), layered coke and ore pellet (**b**) and mixed coke and ore pellet (**c**).

*3.3. The Boundary of the Plug Flow and Convergence Flow in BF*

Due to the significant influence of the solid flow patterns on the smooth operation of BFs, the boundary between the plug flow and convergence flow in BF was determined. As shown in Figure 5a, ten equidistant annular regions (1–10) at different heights along the same section were selected to calculate the regional mean axial velocity of burden. It should be noted that these ten regions are selected from the center of the furnace to the furnace wall, and the data shown in Figure 5a was determined by averaging the values obtained from the three types of the solid flow in Figure 4. Moreover, it can be found that the error bars became large at the lower part of the BF model, i.e., the velocity fluctuations occurred in regions 7–10 at heights of 20–70 mm.

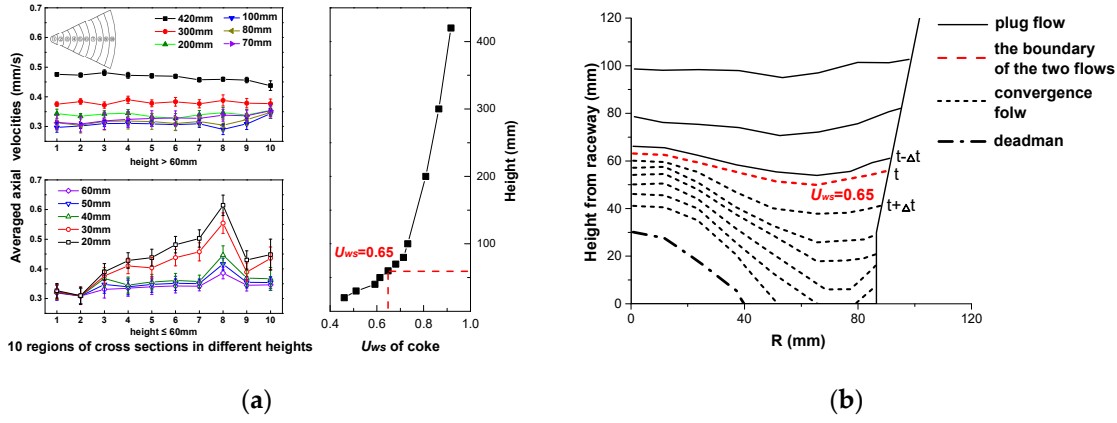

**Figure 5.** (**a**) Averaged axial velocity in 10 regions at different heights of the BF, and the ($U_{ws}$) distributions along the height of the BF. (**b**) The boundary between the plug flow and convergence flow in the BF, which was determined by $U_{ws}$.

Undoubtedly, in an actual production of a BF, the burden timeline in bosh is affected by various factors, such as the cohesive zone and tuyere position. Flow uniformity is an important evaluation index for solid flow. A favorable explanation of the relationship between the appearance of the convergence flow and the differentiation of the particle velocity is necessary. In the paper, based on

standard deviation, the Wilcox–Swailes coefficient (i.e., $U_{ws}$) [30–32] was introduced to evaluate the characteristics of the solid flow, which can be expressed as,

$$U_{ws} = 1 - \frac{\sqrt{\frac{1}{n}\sum_{i=1}^{n}(v_i - \overline{v})^2}}{\overline{v}} \tag{4}$$

where $v_i$ is the local velocity of the labeled particle in m/s, $\overline{v}$ is the average velocity of all particles on the measuring cross section in m/s, and $n$ is the number of particles.

Generally, the velocity stability is favorable when $U_{ws}$ is large, and the maximum value of $U_{ws}$ is 1. As shown in Figure 5a, due to the large $U_{ws}$, it can be found that the solid flow at the top region of the BF model may be uniform. Meanwhile, the results indicate that, with the burden descending, the $U_{ws}$ decreases drastically in the lower part of the bosh. By combining the regional velocity standard deviation with the $U_{ws}$ of the different working conditions, it can be found that the velocity in the vertical direction changes suddenly at a height of approximately 40–80 mm, which corresponds to $U_{ws} \leq 0.7$, as shown in Figure 5a. In addition, Figure 5b shows the timelines in the lower part of the bosh. Evidently, with the burden descending, the plug flow changes to the convergence flow. In detail, the timeline with a $U_{ws}$ of 0.65 is expressed by a red dotted line. The previous time interval ($t - \Delta t$) and the next time interval ($t + \Delta t$) are above and below this red dotted line, respectively. It can be found that a roughly equal interval occurred between the timeline $t - \Delta t$ and the timeline t. However, there is a significant difference between the timeline $t + \Delta t$ and the timeline t. Therefore, the $U_{ws}$ with the value of 0.65 can be defined as a critical value to determine the solid flow patterns, i.e., when $U_{ws}$ is higher than 0.65, a plug flow is dominant, whereas, when $U_{ws}$ is lower than 0.65, a convergence flow is dominant. According to the calculation results, the position of the demarcation line for the plug flow and the convergence flow shown in Figure 5b is in agreement with the experimental and simulation results shown in Figures 3 and 4. This method may be used to analyze the motion behavior of large-scale particles in shaft furnaces.

## 4. Conclusions

To eliminate the wall effect on the structure of solid flow in BFs, a cutting burden method was developed to observe the vertical section of the solid flow by inserting a transparent plate into the experimental BF models. By combining the observations for vertical section of solid flow in experimental BF models via cutting with the simulation results by DEM, we found that the plug flow is the main solid flow pattern in the upper and middle zones during burden descending, whereas a slight convergence flow and a deadman zone form at the lower part of the bosh. The quasi-stagnant zone of the mixed charging method is smaller than that of single particle method. The boundary between the plug flow and convergence flow in the BF can be determined by the velocity distribution of burden and Wilcox–Swailes coefficient, i.e., $U_{ws}$. When $U_{ws}$ is higher than 0.65, a plug flow is dominant, whereas when $U_{ws}$ is lower than 0.65, a convergence flow is dominant. The findings may have important implications to understand the structure of the solid flow in BFs.

**Author Contributions:** Data curation, Y.L.; funding acquisition, X.Z.; supervision, J.W.; riting—original draft, Y.L.; writing—review & editing, Z.J. and X.Z. All authors read and approved the manuscript.

**Funding:** This work was supported by the National Key Research and Development Program of China (2018YFB0605903, 2016YFB0601301).

**Conflicts of Interest:** The authors declare no conflict of interest.

## Appendix A

Before the experiment, a verification experiment was carried out to test the reliability of this cutting method. The transparent thin plate in the experiment is made of organic glass, and this material has a certain bending resistance and strength. The thickness of the plate is 4 mm, and the thickness of the cutting part is less than 1.5 mm.

Firstly, a certain height of yellow particles was placed in the BF model, and then the tracer particles were placed on the top of yellow particle layer. The thickness of the trace particles is 1cm, as shown in Figure A1.

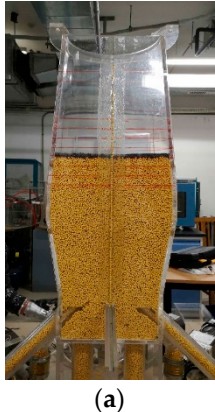

(**a**)

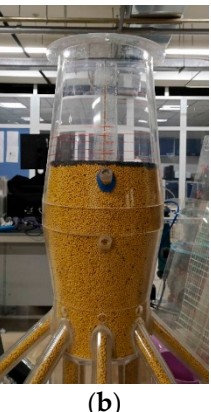

(**b**)

**Figure A1.** Placing the tracer layer. The view of (**a**) the flat wall and (**b**) the semicircular wall.

In the second step, the yellow particles were filled up in the BF body, as shown in Figure A2.

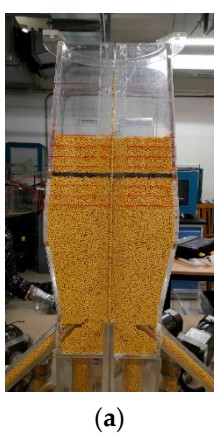

(**a**)

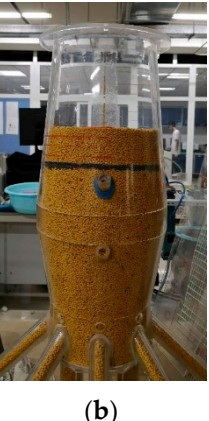

(**b**)

**Figure A2.** Filling up the yellow particles. The view of (**a**) the flat wall and (**b**) the semicircular wall.

For the last step, in the process of the cutting method, the plate was instantly inserted into the burden with a short time. As shown in Figure A3, an experimenter held the BF model, and the other experimenter pushed the transparent thin plate into the burden. The plate slides into the burden along the preset chutes on the upper and lower sides of the BF body. At last, the solid particles on the one side of the transparent plate were removed, and the internal surface was exposed, as shown in Figure A3b.

The average thickness of the tracer particles was 1.133 cm after cutting, as shown in Figure 4b, which is increased by 13% compared with the thickness before cutting, as shown in Figure A4a. Though the shape of the tracing line appears to have a little subsidence phenomenon in the position of the entrance, the shape of the half-circle BF model is a little more than the semicircle. Therefore, the selection of the tracing line on the internal surface needs to remove that part of the entrance, like in Figure A4b. From Figure A4, after the cutting method, no significant changes have taken place to the tracer line, with only slight changes in thickness, so it is considered that this method will not destroy the burden structure.

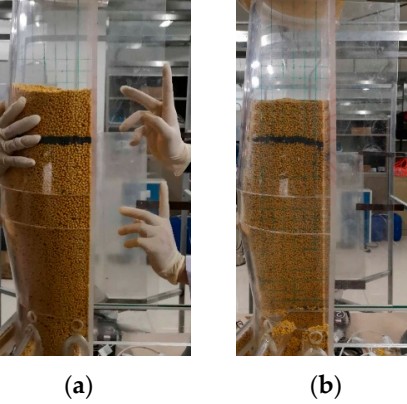

(**a**)                    (**b**)

**Figure A3.** The cutting method in experiment. (**a**) The process of the cutting method; (**b**) the internal surface after the cutting method.

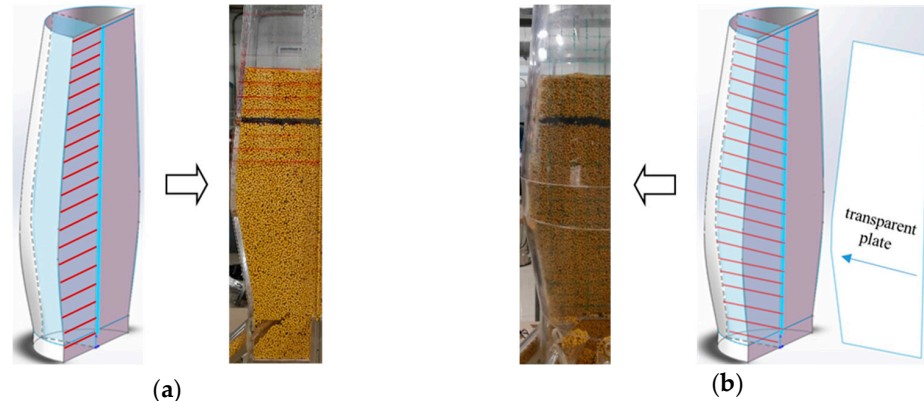

(**a**)                    (**b**)

**Figure A4.** Features of the tracer line before and after the cutting method. (**a**) Tracer line on the flat wall; (**b**) tracer line on the section after the cutting method.

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
