# Peer review of "Vertical Section Observation of the Solid Flow in a Blast Furnace with a Cutting Method"

_metals, doi:10.3390/met9020127_

Round 1
Reviewer 1 Report
The subject developed in this manuscript describes an advanced approach (a cold experimental BF model and numerical simulation via discrete element mode) for following the structure of solid flow in BF (Blast furnace).
As a whole, this is a well-written paper and only some minor corrections are given below.
Yuanxiang Lu 1, Zeyi Jiang 1,2,* and Xinru Zhang 1,3 , Jingsong Wang4, Xinxin Zhang1,2
should be:
Yuanxiang Lu 1, Zeyi Jiang 1,2,*, Xinru Zhang 1,3 , Jingsong Wang4 and Xinxin Zhang1,2
Blast furnace (BF) is a complex metallurgical reactor to produce liquid iron.
“liquid iron” should be “pig iron”.
In fundamental respects,
should be:
In fundamental aspects, (?)
Dong [2]…
should be:
Dong et al. [2]…
Ariyama [3] et al.
should be:
Ariyama et al [3].
“Adema [15] and Ping [16] et al.”
should be:
“Adema et al. [15] and Ping et al. [16]”
“Fu [17]”
should be:
“Fu et al. [17]”
“Xu et al. [18,19]”
should be:
Xu et al. [18] and Yang et al. [19]
“Wright et al.”
should be:
Wright et al. [23]?
Please write “Figure x” instead of “Figure.x”. Check whole text and modify.
Please add a reference for the Hertz–Mindlin model.
Author Response
The subject developed in this manuscript describes an advanced approach (a cold experimental BF model and numerical simulation via discrete element mode) for following the structure of solid flow in BF (Blast furnace).As a whole, this is a well-written paper and only some minor corrections are given below. Response: We really appreciate the reviewer’s recognition about our work. According to the comments, we have carefully revised our manuscript. All the revisions in the text have been clearly marked in the revised manuscript. Point 1: Yuanxiang Lu 1, Zeyi Jiang 1,2,* and Xinru Zhang 1,3 , Jingsong Wang4, Xinxin Zhang1,2 should be: Yuanxiang Lu 1, Zeyi Jiang 1,2,*, Xinru Zhang 1,3 , Jingsong Wang4 and Xinxin Zhang1,2 R: Thanks and implemented. Point 2: Blast furnace (BF) is a complex metallurgical reactor to produce liquid iron. “liquid iron” should be “pig iron”. R: Thanks and implemented. Point 3: In fundamental respects, should be: In fundamental aspects, (?). R: We thank the reviewer for this comment. We have revised “In fundamental respects,…...” on Page 1 (Line 38) to “In fundamental aspects,……” Point 4: Dong [2]…should be: Dong et al. [2]……Ariyama [3] et al. should be: Ariyama et al [3]. “Adema [15] and Ping [16] et al.” should be:“Adema et al. [15] and Ping et al. [16]” “Fu [17]”should be: “Fu et al. [17]” “Xu et al. [18,19]” should be: Xu et al. [18] and Yang et al. [19] “Wright et al.” should be: Wright et al. [23]? R: Thanks and implemented. Point 5: Please write “Figure x” instead of “Figure.x”. Check whole text and modify. R: Thanks and implemented. Point 6: Please add a reference for the Hertz–Mindlin model. R: We thank the reviewer for this comment. We have added a reference to describe the Hertz–Mindlin model on Page 5 (Line180), “The Hertz–Mindlin [29] model including a spring and dashpot…...” [29] Johnson, K.J.P., Cambridge. Contact Mechanics Cambridge Univ. 1985.

Reviewer 2 Report
The authors presented a work for the simulation of the solid flow in a blast furnace. They have prepared a scale model of the BF and have performed simulations with the DEM. The work is interesting and well written. However, some revisions are required before being accepted for publication.
In the introduction, the authors nee to provide some more recent papers that are closely related to the topic of their paper. They should also mention details on the scale model and the numerical methods that were used in the past and incorporated in their work. Furthermore, the authors need to be more specific what is the potential use of the proposed cutting method and the results of the model.
The description of the DEM used in the paper is rather poor; more details are required in order to evaluate the model. The authors need to mention the software used, the properties of the particles and, generally speaking, be more specific on the construction of the model.
Finally, the authors should make some comparisons between their results and similar results found in the literature and mention if the results were anticipated or not.
Author Response
The authors presented a work for the simulation of the solid flow in a blast furnace. They have prepared a scale model of the BF and have performed simulations with the DEM. The work is interesting and well written. However, some revisions are required before being accepted for publication.
Response: We really appreciate the reviewer’s recognition about our work. And thank the reviewer for the valuable comments and suggestions. We have carefully revised the manuscript point by point. All the revisions have been highlighted in the revised manuscript.
Point 1: In the introduction, the authors need to provide some more recent papers that are closely related to the topic of their paper. They should also mention details on the scale model and the numerical methods that were used in the past and incorporated in their work. Furthermore, the authors need to be more specific what is the potential use of the proposed cutting method and the results of the model.
Response 1: We thank the reviewer for raising this suggestion. We have added some new references in the revised manuscript in the Introduction. The details of the main numerical models have been described on Page 2.
Page 1, Line 41 (original Line 40), in the part of Introduction: We have added a new reference and revised “….as reviewed by Yagi [1], Dong et al. [2] and Ariyama et al. [3].” to “….as reviewed by Yagi [1], Dong et al. [2], Ariyama et al. [3] and Kuang et al. [4].”
[4] Kuang, S.B.; Li, Z.Y.; Yu, A.B. Review on modelling and simulation of blast furnace. Steel. Res. Int. 2018, 89, 1700071-1700071/1700025.
Page 2, Line 49-51 (original Line 49): We have added some new references and revised the text “…. molten slag trickle flow [9] .… air pressure drop [14], etc.” to “…. molten slag trickle flow [10, 11] …. the air pressure drop [16], and the flow and wall stress [17], etc.”[11] Yang, W.; Zhou, Z.; Yu, A.; Pinson, D. Particle scale simulation of softening–melting behaviour of multiple layers of particles in a blast furnace cohesive zone. Powder Technol. 2015, 279, 134-145.[17] Samsu, J.; Zhou, Z.; Pinson, D. Flow and wall stress analysis of granular materials around blocks attached to a wall. Powder technol. 2018, 330.
Page 2, Line 54-62 (original Line 53-56): we have added some details to describe the models and revised the text “Adema [15] and Ping [16] et al. evaluated different burden descent models ……to consider the non-uniform descending speed of the burden.” to “Adema et al. [18] compared 3 types of BF geometry (slot models and a pie-slice) with different particle shapes, and concluded that the geometry used should be carefully chosen as it had a very large influence on the solid flow. Ping [19] et al. evaluated different burden descent models under four charging patterns. These models could predict the positions and shapes of different timelines in BF, and the results showed that the C/O charging pattern could influence the shape of cohesive zone and deadman. Fu et al. [20] proposed two models, i.e., geometric profile (GP) model and potential flow (PF) model to consider the non-uniform descending speed of the burden. The model can obtain the descending speed with different C/O ratio and can be applied in the online prediction for the operation of blast furnace.”
Page 8, Line 251-253 (original 228), in the part of Results:
We have added the application of the experimental cutting method and revised the text to “Therefore …. with the literature [26, 27]. The experimental cutting method can solve the problem of the wall effect on the burden descending in shaft furnace, which might have important implication for various industrial applications [25-28].”
We thank the reviewer for raising this concern. Some new references have been updated in the paper which were highlighted. And the details of the main numerical models have been described on Page 2. The application of this experimental method was in Section 3.1 of this paper.
Point 2: The description of the DEM used in the paper is rather poor; more details are required in order to evaluate the model. The authors need to mention the software used, the properties of the particles and, generally speaking, be more specific on the construction of the model.
Response 2: We have added the software and revised “The DEM was adopted to……” (Page 4, Line 167) to “The DEM method was adopted to simulate the solid flow in the experimental BF model using an EDEM commercial software (EDEMTM , England).”
We have added a new paragraph and a new table on Page 5 and 6 (Line 180-200) to describe the details about the DEM method.
“….The Hertz–Mindlin [29] model including a spring and dashpot was used in DEM to calculate the contact forces. The trajectory of a particle is obtained by the equations governing the translational and rotational motions of a particle [14]. In detail, the governing equations for particle i can be expressed as follows:
, (2)
, (3)
where and denote the translational velocity (m/s) and rotational velocity (rad/s) of the particle i, respectively. mi is the particle mass, kg/m3. Ii is the moment of inertia of the particle, kg·m², which is given by . The forces include the gravitational force, ; and the contact forces between the particles and particles-walls. The contact forces and the damping forces in the normal and tangential directions involved are: , , and , N, respectively. The torque acting on particle i involved are: , which causes particle i to rotate by the tangential force, N·m; and , so called the rolling friction torque, which slows down the relative rotation between particles by the normal force, N·m. The forces and torques used in the model are listed in Table 2.
Table 2. Forces and torques acting on particles i.
Forces and torques | Symbols | Equations |
Normal contact force | ||
Normal damping force | ||
Tangential contact force | ||
Tangential damping force | ||
Coulomb friction force | ||
Torque by tangential forces | ||
Rolling friction torque |
Notes: , , , , , , , , , , , , , especially, ,,,,, and mean the equivalent Young’s modulus, normal amount of overlap, equivalent mass, equivalent radius of the particles, coefficient of restitution, equivalent shear modulus, and tangential stiffness of particles, respectively.”
Accordingly, we have revised the Table 3.
Table 3. Parameters of the particles in the DEM simulation.
Parameters | Sector model 1 | Sector model 2 |
Particle shape | Spherical | Spherical |
Particle motion state | Moving bed | Moving bed |
Particle diameter, mm | 2.5 (c), 1.25 (o) | 2.5 (polyethylene) |
Particle density, kg/m3 | 1100 (c), 4000 (o) | 910 (polyethylene) |
Wall density, kg/m3 | 7600 (furnace wall) | 1200 (acrylic) |
Time step, s | 1×10−4 | 1×10−4 |
Total number | Variable | 35000 |
Poisson’s ratio | 0.21 (c), 0.24 (o) | 0.49 (polyethylene) |
Shear modulus, Pa | 1e+07 | 1e+07 |
Coefficient of restitution | 0.3 | 0.3 |
Coefficient of interparticle static friction | 0.63 (c-c), 0.4 (c-o), 0.32 (o-o) | 0.21 |
Coefficient of interparticle rolling friction | 0.05 | 0.05 |
Coefficient of static friction (p - wall A) | 0 | 0 |
Coefficient of static friction (p - wall B) | 0 | 0 |
Coefficient of static friction (p - wall C) | 0 | 0.156 |
Coefficient of static friction (p - wall D) | 0.56 (c-w), 0.31 (o-w) | 0.156 |
Coefficient of rolling friction (p - w) | 0.05 | 0.05 |
(c: coke particle; o: ore pellet; p: particle; w: wall)
Point 3: Finally, the authors should make some comparisons between their results and similar results found in the literature and mention if the results were anticipated or not.
Response 3: We thank the reviewer for this valuable comment. We have added the comparisons between the results of literatures, and the similarities and differences in the literatures have been compared and explained in this part as follows:
Page 7, Line 231-235 (original Line 214)
“Compared with literature [26], there is a longer bosh in this work. The result of the flat wall shows a larger quasi-stagnation zone in bosh, which illustrates that the shape and length of the bosh can affect the shape of this quasi-stagnation zone. In the hearth region, the experimental result in Figure c1 shows that more inclined lines exist in this region, which is different from the results in literature (the curved lines) [26].”
Page 9, Line 279-283 (original Line 228)
The comparison between the results in this article (exp. and sim.) and the literatures have been revised as follows, and this part has been shown in Response 1. “Therefore, during analyzing the heat-mass transfer and metallurgical reactions in the BF, the plug flow should be used to model the solid phase, which agrees well with the literature [30]. The results can provide important information for building the one-dimensional model and the two-dimensional model used to simulate the movement and reaction of coke/ore in BF.”
Page 10 (Line 321-324)
We have added a new description to compare the mathematical means (Uws) with the results by experiments and simulations, “According to the calculation results, the position of the demarcation line for the plug flow and the convergence flow shown in Figure 5 (b) is in agreement with the experimental and simulation results shown in Figure 3 and Figure 4. This method may be used to analyze the motion behavior of large-scale particles in shaft furnace.

Round 2
Reviewer 2 Report
The authors have improved their paper according to the reviewer's comments.
The paper can be accepted in its present form.